# Northern preference for terrestrial electromagnetic energy input from space weather

I. P. Pakhotin [1✉], I. R. Mann [1], K. Xie[1], J. K. Burchill[2] & D. J. Knudsen[2]

Terrestrial space weather involves the transfer of energy and momentum from the solar wind into geospace. Despite recently discovered seasonal asymmetries between auroral forms and the intensity of emissions between northern and southern hemispheres, seasonally averaged energy input into the ionosphere is still generally considered to be symmetric. Here we show, using Swarm satellite data, a preference for electromagnetic energy input at 450 km altitude into the northern hemisphere, on both the dayside and the nightside, when averaged over season. We propose that this is explained by the offset of the magnetic dipole away from Earth's center. This introduces a larger separation between the magnetic pole and rotation axis in the south, creating different relative solar illumination of northern and southern auroral zones, resulting in changes to the strength of reflection of incident Alfvén waves from the ionosphere. Our study reveals an important asymmetry in seasonally averaged electromagnetic energy input to the atmosphere. Based on observed lower Poynting flux on the nightside this asymmetry may also exist for auroral emissions. Similar offsets may drive asymmetric energy input, and potentially aurora, on other planets.

[1] Department of Physics, University of Alberta, Edmonton, Alberta, Canada. [2] University of Calgary, Calgary, Alberta, Canada. ✉email: pakhotin@ualberta.ca

Solar-terrestrial coupling involves energy transfer from the magnetosphere into the ionosphere and atmosphere below. A critical component of this magnetosphere–ionosphere coupling (MIC) involves large-scale field-aligned currents (FACs) which flow in patterns of upwards and downwards sheets in response to solar wind forcing[1,2] and which are related to convection plasma flows in the magnetosphere arising from coupling to the solar wind through magnetic reconnection[3]. Such FACs are established and change dynamically as a result of the field-aligned propagation of Alfvén waves[4]. Such waves are also linked with the formation of some types of auroral features[5,6].

Recent research has addressed the question of whether the aurora are symmetric between the northern and southern hemispheres. For example, the aurora in each hemisphere can be differentially distorted as due to non-zero dawn–dusk component of the interplanetary magnetic field (IMF)[7]. Evidence for a seasonal dependence in the aurora was also presented by Laundal and Østgaard[8] and Østgaard et al.[9]. Asymmetries in the aurora may also occur as a result of differential solar illumination[10], from potential interhemispheric differences in ionosphere–thermosphere coupling as due to the offset of the magnetic dipole from the Earth's centre, as well as from higher-order multipole terms[11].

These studies demonstrate that the auroral forms and their intensities in the two hemispheres can be asymmetric. However, a systematic study of asymmetries in the incoming Poynting flux from electromagnetic plasma waves has not been completed. Recent work found that FACs in the auroral zone tend to be stronger in the north when averaged over a year[12]. The ionospheric conductance is known to have a strong influence on the strength of the FACs[12]. However, in order to assess in situ electromagnetic energy transfer, one requires both electric and magnetic field measurements in order to compute the Poynting vector and this has heretofore not been analysed in detail. Under common assumptions, the magnetic-field-aligned component of the Poynting vector is equal to the height-integrated Joule dissipation below the satellite[13].

In this work, we use data from the European Space Agency (ESA) Swarm mission[14] to assess the seasonal dependence of the electromagnetic energy input associated with MIC at Swarm altitudes, and thereby assess the response of space weather in geospace to solar wind forcing. Preliminary statistics[15] demonstrated a northern preference for electromagnetic energy input during the northern summer. As that study only considered northern summer months, they were unable to assess whether such asymmetry would reverse 6 months later, nor whether there was any systematic seasonably averaged interhemispheric asymmetry.

Here, we show using data from the Swarm satellite, in polar low-Earth orbit (LEO) at an altitude of around 450 km, that, in contrast to the standard paradigm of interhemispheric symmetry, there is persistently higher electromagnetic energy input in the northern hemisphere even when averaged over season. This preference for stronger northern electromagnetic energy input is observed in both the dayside and nightside. Indeed, on the nightside there is a dominance of energy transfer into the north in both near-summer and near-winter solstice seasons.

## Results

Here we examine the electromagnetic energy input into the ionosphere by assessing the Poynting vector associated with perturbations along the satellite world-line, calculated using

$$\mathbf{S} = \frac{1}{\mu_0} \mathbf{E} \times \mathbf{B}. \qquad (1)$$

In Eq. (1), $\mu_0$ is the magnetic constant, and $\mathbf{E}$ and $\mathbf{B}$ denote the electric and magnetic field vectors of the perturbation fields, respectively. By applying band-pass filters, it is possible to remove the influence of large-scale variations of the Earth's main field as measured along the trajectory of the moving satellite, and to focus on the Poynting flux contributions arising from perturbations at various scales of interest. With a single satellite, it is impossible to uniquely separate the impacts arising from spatial and temporal variations along the satellite world-line. However, as shown for example by Knudsen et al. and Pakhotin et al.[16,17], analysis of the wave impedance as a function of frequency in the Swarm frame provides strong evidence for the importance of Alfvén waves in MIC.

Figure 1 shows the statistical Poynting flux over four separate 1-month-long time periods, one in the northern near-summer solstice conditions (1–31 July, 2016; Fig. 1a, b), two around the spring equinox period (one in late northern spring (20 March–20 April, 2017; Fig. 1c, d) and one 20 February–25 March, 2016; Fig. 1e, f), and one in the northern near-winter solstice conditions (15 November–15 December, 2016; Fig. 1g, h), for both the dayside (Fig. 1a, c, e, g) and the nightside (Fig. 1b, d, f, h) as determined by magnetic local time (MLT). These intervals were chosen to reflect periods where the Swarm A orbits were in similar noon–midnight local time orientations. In Fig. 1, the error bars show the median spanned by the upper and lower quartiles in the statistics, with the scale dependence of the Poynting flux as a function of frequency derived by the application of a time-domain Savitzky-Golay low-pass filter of varying width along the x-axis (see "Methods" for details). It can be seen that on the dayside during near-summer solstice, there is a clear statistical preference for more electromagnetic energy to be driven into the northern hemisphere than the southern hemisphere at Swarm altitudes. On the dayside during near-winter solstice, the preferential direction of the energy transfer does reverse such that there is more Poynting flux directed into the southern hemisphere. However, and very significantly, the asymmetry in the interhemispheric energy transfer is much smaller than in the near-summer solstice period. As a result, there is a clear preference for more energy transfer into the north. Indeed, if the results from these 2 months approximating the near-summer and near-winter solstice periods are summed, the implied summer–winter seasonally averaged Poynting flux will have a clear net northern preference.

On the nightside (Fig. 1b, d, f and h), the northern preference for electromagnetic energy transfer is even more stark. During the near-winter solstice on the nightside (Fig. 1h), there is a reduction in the northern preference, but the northern preference in Poynting flux appears to persist even near the northern winter solstice on the nightside.

In all cases, the error bars plotted in Fig. 1, and which refer to the 25% and 75% quartiles in Poynting flux, appear to be a significant fraction of the median. However, we emphasise that this feature should not be interpreted as a low statistical significance of our result demonstrating northern preference for electromagnetic energy transfer seen at Swarm. Instead, the large range of mean Poynting flux magnitudes represented by the quartiles simply reflects the expected large variability in the magnitudes of energy flux from hour-to-hour and day-to-day in response to non-steady solar wind forcing. Supplementary Figs. 1 and 2, for the northern near-winter solstice period, as well as Supplementary Figs. 7 and 8, for the near-summer solstice period, respectively, show this effect clearly. It can be seen that during more intense geomagnetic activity, the magnitude of the Poynting flux increases in both hemispheres (e.g., Supplementary Figs. 2a, b and 8a, b). This can be seen particularly during conjugate observations from adjacent northern and southern hemisphere passes, where the Poynting flux

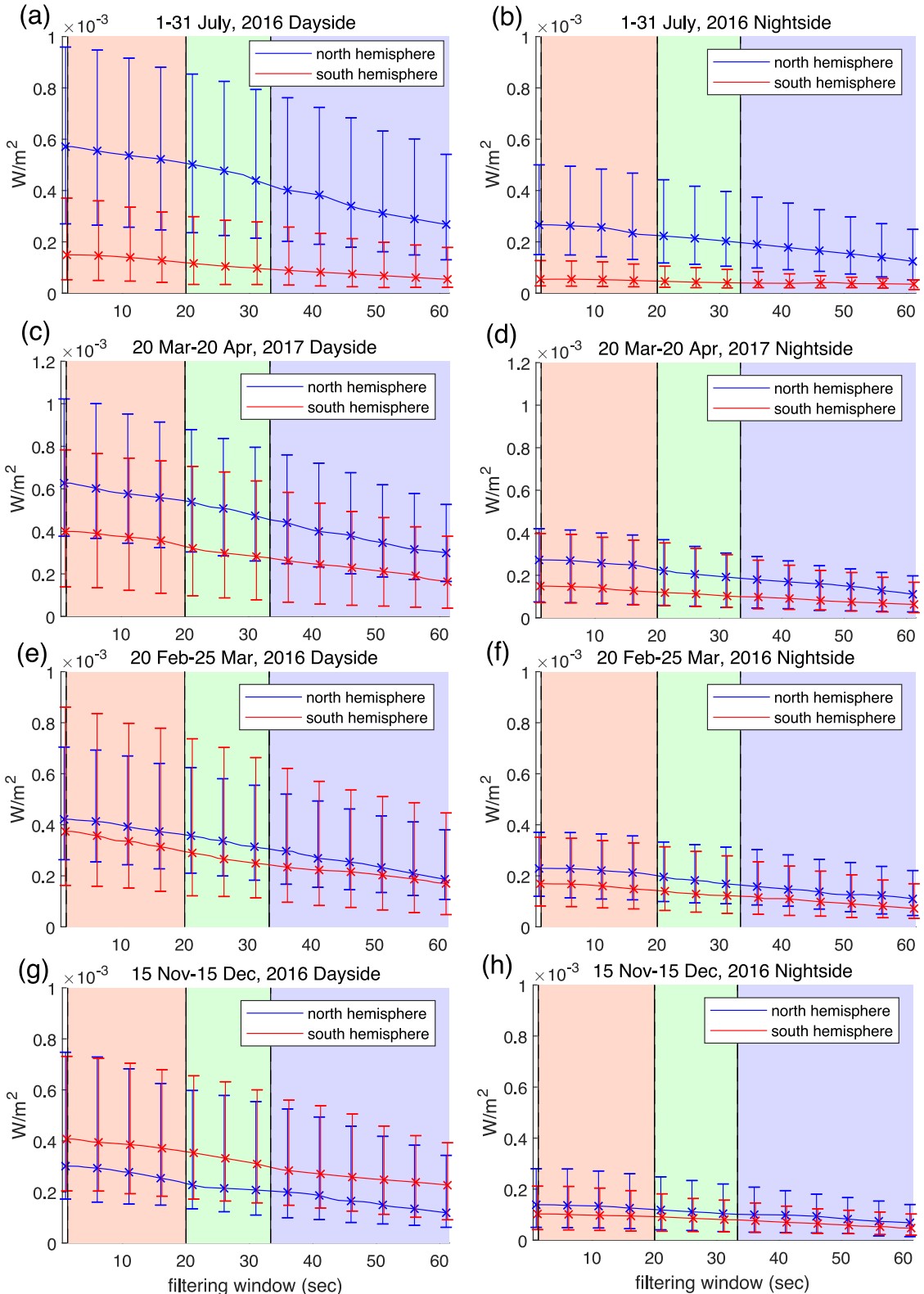

**Fig. 1 Mean Poynting fluxes observed by Swarm A satellite crossings of the northern and southern auroral zones.** Dayside (**a**, **c**, **e**, **g**) and nightside (**b**, **d**, **f**, **h**) auroral zone crossings for northern near-summer solstice (1–31 July, 2016; **a** and **b**), northern spring period (20 March–20 April, 2017; **c** and **d**), northern equinox period (20 February–25 March, 2016; **e** and **f**) and northern near-winter solstice (15 November–15 December, 2016; **g** and **h**), during periods where the orbits are confined to the noon–midnight local time sector (see also Supplementary Fig. 9). The panels compare Poynting flux as a function of transverse spatial scale, derived using a time-domain Savitzky-Golay filter, in the northern and southern hemispheres (blue and red, respectively) and plot the median Poynting flux power values (solid line) and the 25% and 75% quartiles (error bars). Three regions separating small-scale (10–150 km), mesoscale (150–250 km) and large-scale (>250 km) phenomena as observed along the Swarm orbit, as per the definitions in ref. [37] are denoted by pink, green and blue backgrounds, respectively.

is seen to increase and decrease in tandem in both hemispheres in response to varying levels of driving. In particular for the near-summer solstice period, it can be seen that the northern Poynting flux dominates over the conjugate southern hemisphere counter-part, both on the dayside and on the nightside, in the time domain across the whole interval despite it spanning a wide range of intensities of solar wind driving conditions. Therefore, the northern preference for electromagnetic energy input persists in the time domain from event to event, and not just when combined statisti-cally as in Fig. 1.

Figure 1 further shows that in general the electromagnetic Poynting flux observed at Swarm appears to be smaller on the nightside than on the dayside. This is evident both in the near-summer and near-winter solstice periods and appears to be a general characteristic of the magnitude of the observed electro-magnetic energy input arising from electromagnetic fluctuations at this altitude. A likely explanation for this is that a significant fraction of the incoming electromagnetic energy is converted to the kinetic energy of downgoing auroral electrons as a result of coupling at higher altitudes above Swarm in the nightside auroral acceleration region (AAR) located around 4000–12,000 km in altitude[18]. This inference is consistent with the concept whereby the ionospheric feedback instability[19] can produce discrete arcs which convert incoming electromagnetic energy into field-aligned electron acceleration in the AAR. This feedback process happens preferentially at night where the background conductivity is low, and where in the absence of dayside EUV illumination strong conductivity gradients can be formed[20]. In such a paradigm, the reduction in nightside Poynting flux observed at Swarm, located below the AAR, may be explained as a result of significant energy removed in association with discrete arc auroral electron acceleration above.

Interestingly, in the data shown in Fig. 1, the interhemispheric energy fraction appears to be independent of scale. This suggests that the processes responsible for the observed asymmetry are most likely self-similarly active at and/or self-similarly impact all transverse scales considered.

The time periods around the equinoxes also show similar behaviour in terms of the northern preference for electromagnetic energy transport at Swarm altitudes, with the time periods closer to the equinoxes exhibiting behaviour that falls between the dynamics seen near the solstices (see Fig. 2). The median and quartile Poynting fluxes at small, medium and large spatial scales shown in Fig. 2 continue to show the northern preference, and how this preference evolves with season from the peak inter-hemispheric asymmetry in the northern near-summer solstice, through the equinoxes, to the northern near-winter solstice. Interestingly, the behaviour of the interhemispheric asymmetry in electromagnetic energy flux is self-similar at small, medium and large scales, suggesting that most likely the same physical pro-cesses are active in MIC across the entire range of spatial and temporal scales shown in Figs. 1 and 2, and during all seasons. It is also possible that energy is transferred between scales within this system via a cascade[21]. It can be seen that the sum totals of Poynting fluxes (north hemisphere flux plus south hemisphere flux) remain relatively similar across the seasons on the dayside (as denoted by the pink plots in Fig. 2a, c, e), suggesting a rela-tively constant median total energy input into both hemispheres is then re-distributed differentially into the two hemispheres—but with a seasonally averaged northern preference which is especially strong near the summer solstice. On the nightside, the same conclusion applies—except that the northern preference is even stronger than on the dayside. Indeed, on the nightside, the northern preference is so pronounced that the median Poynting flux at Swarm altitudes is always much larger in the north, and at all scales, independent of season—even near winter solstice.

Note that in our analysis, we have taken care to exclude the possibility that the northern preference we report could occur as a result of sampling bias, for example as a result of the inclination of the Swarm orbit generating impacts from differentially sam-pling auroral zone crossings at different angles of attack with respect to the auroral oval. For example, the orbits of Swarm A for all four seasonal time periods were chosen to be confined in the noon–midnight meridian (see Supplementary Fig. 9). More-over, exploration of limited subsets of the two time periods in Fig. 1, focusing on only MLT times one hour before or after 00 or 12 MLT (i.e., more strictly confined to the noon–midnight local time meridian) also produced similar results further confirming the characteristics reported in Fig. 1 are real (see e.g., Supple-mentary Fig. 10). Note that although the electromagnetic fields are derived in local coordinates, both perpendicular polarisations of **E** and **B** fields are used to derive the parallel component of the Poynting flux. As such the reported northern preference for electromagnetic energy transfer appears to be geophysical in origin, and not the result of sampling bias or orbital orientation effects.

## Discussion

Here we advance a paradigm which can explain the observed persistent asymmetry and northern preference for incoming Poynting flux at Swarm altitudes based on the known offset of the magnetic dipole moment from the centre of the Earth towards the northwest Pacific[22]. This offset generates different relative effec-tive solar illumination of the auroral ovals in the northern and southern hemispheres arising from the rotation of the Earth. The offset can also introduce asymmetries in the magnetic fields in the auroral zones as well (cf. ref. [10]). A model of the north and south auroral ovals at two particular instants, shown in blue and red, respectively, is shown in Fig. 3a, b. In each panel, the two ovals are over-plotted in a two-dimensional projection of a polar view of the Earth in the geocentric solar ecliptic (GSE) $x$–$y$ plane, and where the $x = 0$ line marks the terminator and with the nightside beyond the terminator shaded in grey. As the Earth rotates, the offset of the dipole axis from the rotation axis sweeps the auroral ovals, whose location is defined by the magnetic field, further into and out of the sunlight; the offset of the dipole from the Earths centre making the excursions into and away from the sunlight more pronounced in the southern hemisphere than the north. This also changes with season. To illustrate this effect, Fig. 3a shows an example from the northern summer solstice, while Fig. 3b shows the northern winter solstice, at the same UT.

The offset of the magnetic dipole from the Earths centre means that in the south the magnetic pole at the Earth's surface is fur-ther from the Earth's rotation axis than the north magnetic pole[10,11] As a result, the southern auroral oval experiences more diurnal variation in its motion both across the terminator into the nightside, and across the terminator into the dayside 12 h later, than its northern counterpart as a result of the rotation of the Earth. This means that at certain times, the southern oval spends fractionally more time in darkness than the northern oval, and at others fractionally more time in daylight. To illustrate the impacts of the Earth's rotation on the extent of the ovals in the $x$–$y$ plane through one Earth rotation, the dashed lines show the circles which mark out 65° MLAT while the solid circles show the 75° MLAT. This MLAT range may be taken to be roughly repre-sentative of an auroral oval. Meanwhile the bold line traces circles which show the locus of the geomagnetic poles (90° MLAT).

In Fig. 3, it can be seen that the maximum area of the northern auroral oval which is in shadow during (northern) summer sol-stice (Fig. 3a) is approximately one-quarter of the total oval area, whereas the maximum shadow of the southern oval during winter

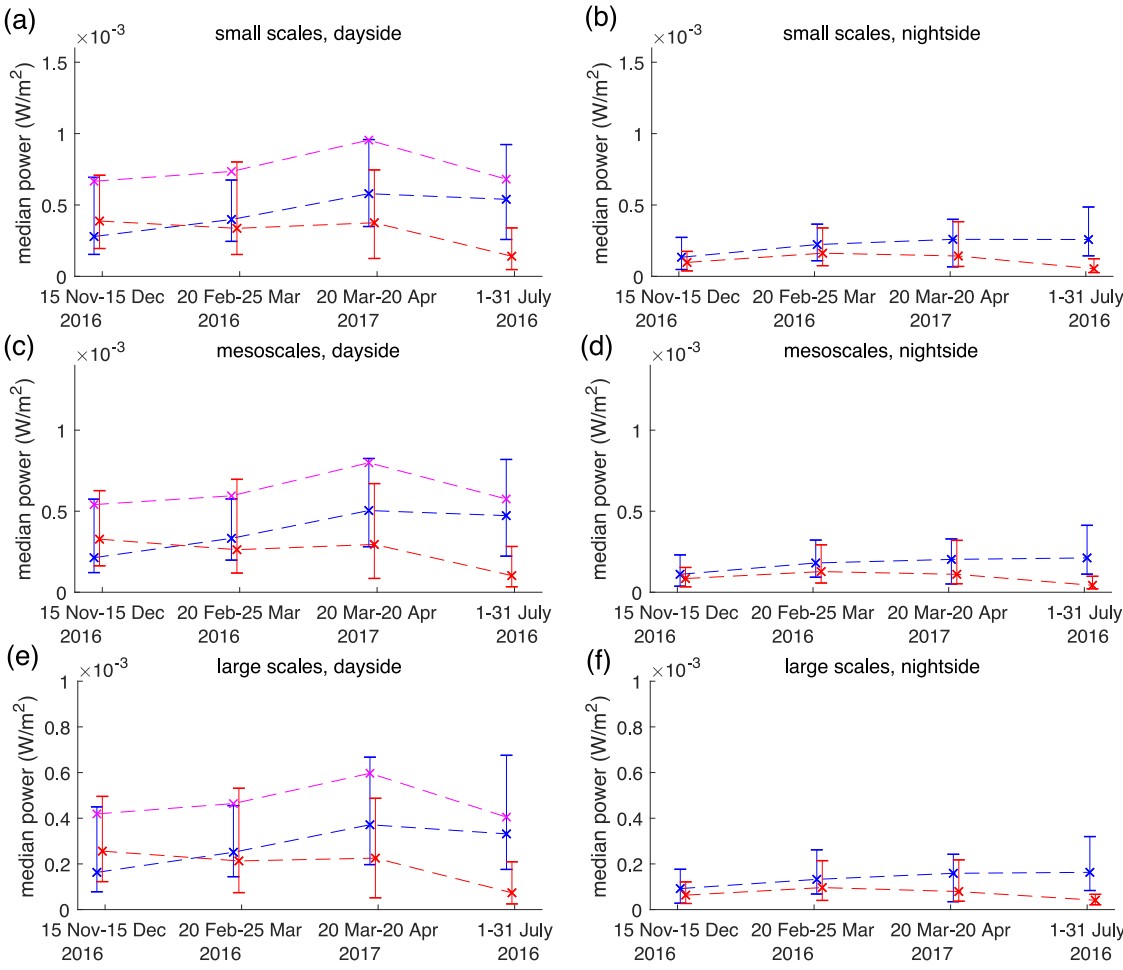

**Fig. 2 Median values of electromagnetic power at small, medium and large scales for near-solstice and near-equinox periods in the noon–midnight local time meridian.** Electromagnetic energy in northern, southern and combined (northern plus southern) dayside (blue, red, pink, respectively; **a**, **c**, **e**) and in the northern and southern hemisphere on the nightside (blue, red, respectively; **b**, **d**, **f**) passes are shown at three spatial scales. Band-pass filtering was done using the Savitzky-Golay filter, with small scales (**a** and **b**) derived using time domain filters characteristic of 75 km along track scales; medium scales (**c** and **d**) characteristic of 200 km along track scales; and large scales (**e** and **f**) characteristic of 350 km along track scales. Dashed lines connect the median values (marked with crosses), while the error bars show the lower and upper quartiles in terms of magnitude. The positions of the medians and error bars for different MLT and hemisphere sectors during the same time periods have been slightly shifted on the x-axis for visual effect and to enable the reader to better differentiate between the error bars. Note as explained in "Methods" that all electromagnetic power values are derived by integrating in time along the spacecraft world-line during an event, and dividing by the duration of the event.

solstice is larger, reaching as much as approximately one-third of its total area. According to the hypothesis described above, discrete auroral acceleration would occur preferentially in dark background ionospheric conductivity conditions. This would make the southern oval more susceptible to losing energy to nightside discrete auroral electron acceleration processes, consistent with the geometrical aspects shown in Fig. 3 and with the nightside reduction in electromagnetic energy transfer observed at Swarm altitudes shown in Figs. 1 and 2.

Meanwhile on the dayside, the southern oval also similarly experiences more variation in solar illumination than the north, potentially traversing further into and dwelling longer in sunlit dayside regions where there is expected to be a greater mismatch between the Pedersen and Alfvén impedances as a result of increased background Pedersen conductance due to dayside solar EUV illumination. This would be expected to lead to greater median Alfvén wave ionospheric reflection coefficients in the south, as per the ionospherically reflecting Alfvén wave paradigm[16,17,23,24]. In turn this could lead to a stronger reflection

of Poynting flux from the southern ionosphere back towards the equator than in the north. This may lead to an overall redirection of a fixed equatorial energy source on the dayside away from the southern hemisphere and into the northern hemisphere, in line with the observations presented in Fig. 2a, c, e. These two dayside and nightside Alfvén wave processes may therefore generate a different MIC response as a result of different ambient dayside and nightside ionospheric conductivity conditions. Nonetheless, they could occur in tandem and could collectively be responsible for the observed northern Poynting flux preference both on the dayside and on the nightside. In both cases, the effect would be to create the observed northern preference for incoming Poynting flux when observed at Swarm altitudes.

The seasonally averaged northern preference for electromagnetic energy input at Swarm altitudes seems to exist in both the dayside and nightside auroral zones. On the nightside, it implies stronger discrete arc auroral precipitation in the south, as residual Poynting flux there is lower and as discrete arcs are expected to be the primary absorber of electromagnetic energy

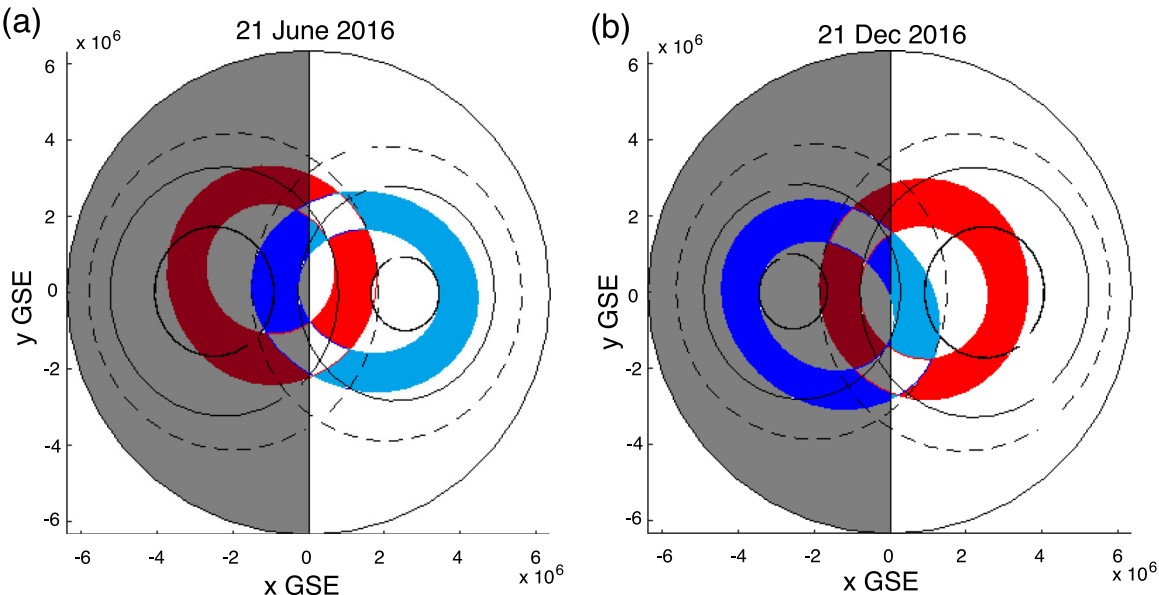

**Fig. 3 Asymmetric interhemispheric auroral zone solar illumination.** The figure shows the two-dimensional projection in GSE *x–y* coordinates of the traces of the north (blue) and south (red) auroral ovals, each assumed here to lie between lines of constant altitude adjusted corrected geomagnetic (AACGM) latitude from 65-75° during matching UT times for northern summer (**a**; 21 June 2016) and northern winter (**b**; 21 December 2016) solstices. The solid and dashed black lines denote the loci of the edges of the ovals in the northern and southern hemispheres as a result of diurnal rotation during the northern summer (**a**) and winter (**b**) solstices. The bold solid black circle at high latitudes denotes the locus of the geomagnetic poles. The solid vertical line denotes GSE *x* = 0 mark/the terminator. The area in Earth's shadow is shown in grey.

incident from above. This conclusion is also consistent with some previous conjugate auroral observations at specific instants in time[8] but which has not been examined statistically in terms of electromagnetic energy input. Meanwhile on the dayside, our results appear to complement the statistics of Newell et al.[25] who used DMSP data to observe auroral electron energy flux as a function of season and hemisphere (note, however, that Newell et al.[25] did not explicitly look at north–south seasonal asymmetries as we do here).

For example, Newell et al.[25] reported that all types of aurora maximised on the nightside in local winter. Meanwhile, they observed that the electron fluxes were higher on the dayside in local summer. The former observation is consistent with our hypothesis that the electromagnetic energy we observe on the nightside is lower during local winter than summer, which we propose occurs as a result of preferential discrete auroral electron acceleration in the AAR above Swarm. The fact that we also observe higher Poynting fluxes in the dayside summer hemisphere suggests that both auroral acceleration and the residual Poynting flux penetrating to Swarm altitudes under the AAR are higher in local summer than local winter on the dayside. Based on our results, we hence suggest that dayside discrete aurora might also be more pronounced during the northern summer than southern summer, an assertion that could be verified by a future study looking for this potential seasonal interhemispheric asymmetry using DMSP particle data. Similar asymmetries in seasonal interhemispheric nightside discrete auroral electron flux could also be investigated using DMSP data in the same way.

On the dayside, we also observe that the sum of northern and southern Poynting flux appears to be similar across seasons. Certainly there is a preference for more energy input into the summer hemisphere, but the northern preference means that this is more pronounced in the northern summer rather than the southern summer on the dayside. Overall, this suggests that the total dayside electromagnetic energy which reaches Swarm altitudes might be rather constant with season, but that ionospheric

conductivity effects and asymmetric Alfvén wave reflections result in a redistribution of the incoming energy flux from one hemisphere to the other. This favours the summer hemisphere, with a more pronounced interhemispheric asymmetry occurring during the northern summer. Interestingly, the results of Hatch et al.[26], using statistics obtained from the FAST satellite, which traverses a range of heights above Swarm, also lend some support to the above hypothesis.

As shown in Figs. 1 and 2, these same dayside and nightside interhemispheric seasonal asymmetries occur across a wide range of spatial scales, and in general the same behaviour is maintained across small (10–150 km), medium (150–250 km) and large (>250 km) spatial scales. Previous work has shown that to some degree, similar behaviour can be inferred in terms of FACs[12] which must logically be associated with these electromagnetic perturbations as well—especially since they are related to Alfvén wave dynamics. For example, previous observational studies have made the link between Alfvénic disturbances and FACs, and have suggested that both may be explained as part of the same physical framework which can explain the characteristics of MIC at these scales[17,24,27,28]. This study is the first to demonstrate that, when seasonally averaged, the high-latitude electromagnetic Poynting flux observed across a wide range of scales has a definitive northern preference both on the dayside and on the nightside. The incoming electromagnetic energy, and indeed likely the incoming discrete auroral electrons, are expected to additionally play a role in driving ionosphere–thermosphere–atmosphere (I–T–A) coupling below, perhaps including effects from gravity waves and other atmospheric phenomena, driven from above (see e.g., ref. [29]). Therefore, the northern preference for electromagnetic energy input which we report here could also be very important in relation to impacting the dynamics of the global coupled magnetosphere–ionosphere–thermosphere system.

Overall, we propose that, for electromagnetic energy input at Swarm altitudes, northern preference can likely be explained by the relative displacement of the north and south auroral ovals

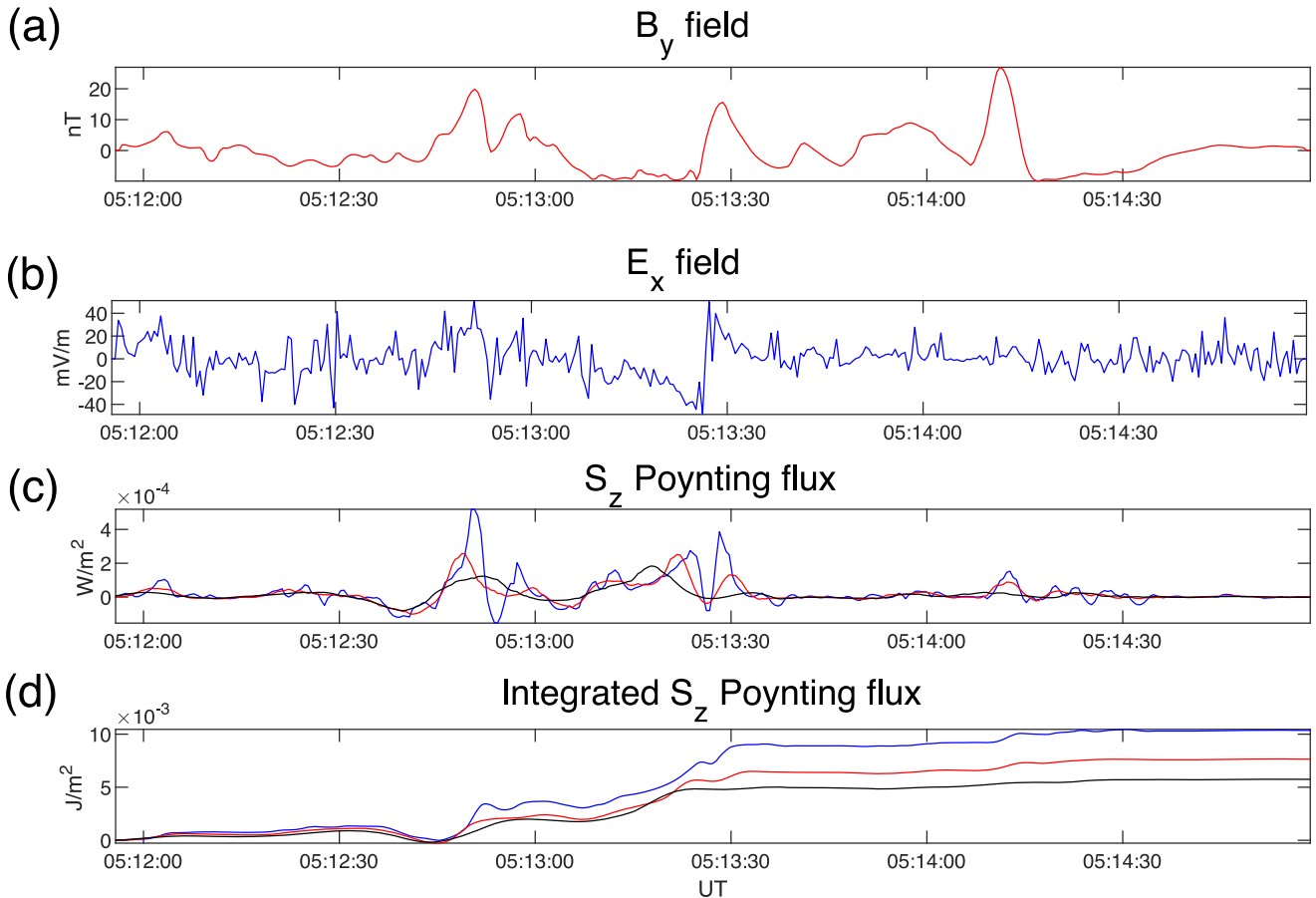

**Fig. 4 Example event selection during a Swarm A auroral zone crossing.** Magnetic and electric field and related field-aligned electromagnetic energy from an event selected by the analysis algorithm between 05:12:00 and 05:15:00 UT on 17 November 2016. Panels show **a** magnetic field $B_y$ component, **b** electric field $E_x$ component, **c** $S_z$ component Poynting flux low-pass filtered with S2avitsky-Golay filter of window sizes 9, 27 and 47 s (blue, red and black, respectively), capturing small-scale, mesoscale and large-scale perturbations, and **d** along-track integrated Poynting flux using the three filtered time series from **c**. Note that the event is selected based implicitly on coherent electromagnetic fields, and with its temporal extent defined from the local peak in the low-pass filtered Poynting flux to a specified lower threshold (see text for details). Please note that error estimates on the electric field are not provided in the dataset. Please refer to ref. [36] for a discussion of the errors.

with respect to the Earth's rotation axis, causing effective inter-hemispheric differential solar illumination of the two auroral ovals. This effect may also be present on other magnetised planets or moons where the magnetic dipole is offset from the planet's centre, asymmetries in MIC occurring as a result of the impacts of differential ionospheric conductivity.

## Methods

The ESA Swarm mission, launched in late 2013, comprises three identical space-craft equipped for making simultaneous, high accuracy and high cadence magnetic and electric field measurements. This study uses data from the Swarm A satellite, in a ~450 km polar orbit, equipped with the fluxgate magnetometer instrument[14] measuring magnetic field vectors at 50 samples/s and the Electric Field Instrument[30] measuring ion velocity vectors at 16 samples/s based on observed ion distributions from two sensors, which are converted into a 2 Hz electric field data product. Under the assumption of frozen flux of the observed ions based on ideal Ohm's Law, where:

$$\mathbf{E} = -\mathbf{v} \times \mathbf{B}. \qquad (2)$$

In Eq. (2), the inferred velocity vectors (**v**) can be converted into electric field vectors in the plane perpendicular to the magnetic field.

The automatic event identification algorithm used here has been designed to extract useful scientific data from synchronous magnetic and electric field measurements while addressing known caveats in the electric field data (e.g., uncertainties regarding offsets and magnitudes of the electric field instrument data[31,32]). It is based on the methodology employed by Park et al.[33]. Only magnetic latitudes of between ±60° and ±80° magnetic latitude are considered for the analysis. This avoids low-latitude phenomena such as plasma bubbles[34] and localised extreme

high-latitude phenomena such as polar cap traversals[35]. The magnetic and electric field data are rotated into the mean-field aligned (MFA) frame, where the z-axis points along the direction of the mean magnetic field, the x-axis points towards the geomagnetic North Pole, and the y-axis completes the triad facing eastwards. A sliding 3-min window is used for mean field calculation.

For event detection and selection, the Poynting flux is calculated by crossing the electric and magnetic field time series after applying a second-order Savitzky-Golay filter with window size of 60.5 s, to remove any residual mean field influences or large-scale electric fields, as well as any uncertainties with the electric field baselines which are a known artefact in the electric field data. The modulus of this Poynting flux is then low-pass filtered with a 120.5 s moving mean filter to obtain its magnitude envelope. Where the magnitude of this Poynting flux envelope exceeds an empirically determined threshold, the event is flagged and event duration defined both backwards and forwards in time until the magnitude of the Poynting flux envelope drops below a second, lower, empirically determined threshold. For the analysis presented here, the thresholds are 25 and 8.75, respectively. This time window then defines a single event. Only electric field datasets flagged with the quality flags 1 ("use in consultation with EFI TII team at University of Calgary")[36] are used in the analysis.

The 2 Hz electric field estimates are provided for both the horizontal and the vertical sensors. Based on the caveats described in ref. [36], the mean from both TII sensors is utilised. Since the analysis in this study used high-pass filtering to focus on relatively small scales, it is deemed acceptable to use the full 3-D vector for Poynting flux calculations. A separate test performed using only the along-track component of the electric field, which is identical in both sensors, to calculate the Poynting flux, also reproduced the observed northern preference.

The selected events must all occur at locations between 60° and 80° magnetic latitude and event length must be at least 150 s long. The time series are extended with zeros for 30 s at the beginning and the end, which serves as padding to allow all filter sizes to fully capture the energy content in each event

without edge-effect distortion. For the selected events, the electric and magnetic field data are then passed through a Savitzky-Golay filter of second order and of various window sizes, from 1 (no effective high-pass filter) to 60.5 s, at 0.5-s intervals. The cross product of the two band-passed signals gives the Poynting flux in that frequency band. The Poynting flux is integrated over time along the spacecraft trajectory to obtain the integrated apparent energy flow through the satellite world-line as it crosses the event region. This is repeated for the entire range of low pass filter window sizes for each event. Mean Poynting fluxes for each event are obtained by dividing the integrated Poynting flux values by the event duration. The median and quartiles are obtained for these Poynting flux values from all of the events flagged by the algorithm.

A representative example of the analysis is shown in Fig. 4 for an auroral zone Swarm A crossing event from 05:12:00 to 05:15:00 UT on 17 November 2016, flagged by the algorithm. It can be seen that there is good correspondence between the magnetic field (Fig. 4a) and electric field (Fig. 4b) data, suggesting mostly positive (downwards) Poynting flux throughout. This is evidenced in Fig. 4c and d where the (parallel) Poynting flux remains largely positive. The Poynting flux shown in Fig. 4c is plotted for three Savitsky-Golay filter windows—9 s (blue) for small-scale phenomena, 27 s (red) for mesoscales and 47 s (black) for perturbations associated with larger scales. It can be seen that the Poynting flux reduces as progressively more low-pass filtering is applied to the constituent electric and magnetic field time series. Figure 4d shows the time integrals of the fluxes in Fig. 4c demonstrating the accumulation of Poynting flux on the satellite's world-line as it crosses the perturbation region. It can be seen that the cumulative Poynting flux passing through the satellite during the event is approximately half the value for large scales as for meso- and small-scales, suggesting that large-scale perturbations associated with global FAC systems carry only half of the electromagnetic energy into the ionosphere during this particular event.

## Data availability

The ESA Swarm data can be obtained from the ESA server at swarm-diss.eo.esa.int. The Swarm A 2 Hz magnetic and electric field data are accessible at swarm-diss.eo.esa.int/ Advanced/Plasma_Data/2Hz_TII_Cross-track_Dataset/New_baseline/Sat_A (browser access: https://swarm-diss.eo.esa.int/#swarm%2FAdvanced%2FPlasma_Data% 2F2Hz_TII_Cross-track_Dataset%2FNew_baseline%2FSat_A). Geomagnetic conditions and L1 information (IMF Bx, IMF By, IMF Bz, AE index, AL index) can be found at https://omniweb.gsfc.nasa.gov/form/omni_min.html.

## Code availability

Standard statistical and data analysis methods were used to generate the results in this manuscript; no custom code is required beyond that outlined in the "Methods" section. Full code can be provided upon request. Coordinate conversion to magnetic coordinates is performed using the IRBEM 4.4.0 library which may be available at https://craterre. onera.fr/prbem/irbem/description.html.

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

## Acknowledgements

I.P.P. is supported by the European Space Agency Living Planet Fellowship "Swarm Investigation of the Energetics of Magnetosphere-Ionosphere Coupling (SIEMIC)". The view expressed in this publication can in no way be taken to reflect the official opinion of

the European Space Agency. I.P.P. is also supported in part by the Canadian Space Agency Class Grant "What role do Alfvén waves play in energy transfer in the dynamical magnetosphere-ionosphere system?" I.R.M. is supported by a Discovery Grant from Canadian Natural Sciences and Engineering Research Council (NSERC). Swarm is a European Space Agency (ESA) mission with contributions from the Canadian Space Agency (CSA) for scientific analysis of Electric Field Instrument data. D.J.K. and J.K.B. are supported by grants from the CSA and ESA.

## Author contributions

K.X. and I.P.P. developed the software algorithm for Poynting flux analysis. I.R.M. provided overall research direction including interpretation of the results, and assisted in software refinement. J.K.B. and D.J.K. provided the electric field datasets and assisted with the interpretation.

## Competing interests

The authors declare no competing interests.
