## [Peer Review File · Nature Communications]

REVIEWER COMMENTS

Reviewer #1 (Remarks to the Author):

Review of Northern Preference for Terrestrial Electromagnetic Energy Input from Space Weather by Pakhotin et al.

The paper presents statistical evidence that incident Poynting flux at low Earth orbit is on average stronger in the northern hemisphere compared to the southern hemisphere. The authors convincingly argue that this intriguing and surprising finding is geophysical, and not an effect of measurement bias - although I have one concern in this regard that I would like to see addressed. The authors also give suggestions for what could cause the observed asymmetry. I struggle with the premise of some of these ideas, and I think there are gaps that should be given some attention before the paper is accepted, which I hope it will be. Apart from this, the writing is generally very clear.

Detailed review

The potential bias from different orbit orientations relative to the auroral zone is mentioned in the paragraph starting on line 158. The authors seem to be most concerned with the effect this has on the calculation of event averages, which clearly is important. However, I wonder if this impact angle also would affect the calculation of Poynting flux itself, since the ion drift measurements from Swarm are (to my knowledge) less accurate in the along-track direction than cross-track. I would think that, if we performed this analysis with only cross-track data, we would find a northern preference simply because the orbit is more often inclined with respect to the oval in the south, due to the larger pole distance (and since the ion drift is typically zonal). If the observed statistical variation is dominated by the more precise cross-track measurements, this bias could be important. I think you could test it in this way: If you run the analysis mentioned on line 344 (with only cross-track drift/along-track E) with a synthetic constant signal for E, and still get a northern preference, the bias influences the results.

In the discussion (paragraphs starting on line 216), I get the impression that the magnetosphere radiates equal amounts of energy to the two hemispheres (e.g., the use of "fixed equatorial energy source"), and that it gets partitioned and reflected in different ways near the ionosphere. For example, it is claimed that since the Southern hemisphere can be more dark, more energy is being diverted into accelerating electrons at the expense of Alfvén wave power, and therefore wave power is stronger in the north. If this is intended, I think the premise of symmetric incident energy should be justified. I see no reason why it should be symmetric to begin with, since different conductivities imply that different amounts of energy are required to bring the ionosphere and

magnetosphere in balance. To first order I would expect more Alfvén waves going back and forth between magnetosphere and ionosphere in the most sunlit hemisphere, simply because that hemisphere is harder to move due to coupling with neutrals. "First order" because particle precipitation and feedback instabilities complicate the picture.

The argument that Alfvén wave power is lost to particles seems to imply a prediction: If you measure Alfvén wave power in or above the AAR, you should get a different result in terms of north-south asymmetries, since the partition has not yet happened. There is at least one paper that has used FAST to statistically analyze Alfvén wave power, separately in the two hemispheres (Hatch et al., 2018, <https://doi.org/10.1016/j.jastp.2017.08.009>). It does not seem to be directly comparable, but it may be worth mentioning.

On line 290, it is claimed that this result can have implications for understanding long-term atmospheric climate response. This seems very speculative, and I recommend removing it unless further justification and citations can be given.

Minor issues:

MIC: I think it would be better to spell out magnetosphere-ionosphere coupling.

line 89: "energy transfer remains slightly northward" - Do you really mean to say that the median direction is northward?

line 91: Missing "to" before "be"

figure 2: I strongly suggest using the same color scheme in the two columns

line 222: remove "of" before "shown"

line 258: The only way I can make sense of this sentence is if the second "both" is removed.

line 358: "these three quantities" which?

Reviewer #2 (Remarks to the Author):

In general I enjoyed this manuscript and feel it would be suitable for publication in Nature with some minor changes, which I enumerate below. My most serious criticism is of the method, particularly lines 347–359; I did not feel this went into enough detail to aid reproducibility.

Science

Line 81--84: This is very similar to the result of [12], which may be worth noting here.

Figure 1: Is there a reason not to show similar pair(s) of panels for the spring and/or autumn equinox?

Lines 96--97: "Supplementary Material Figures 1 and 4, for the northern near-winter and near-summer solstice periods, respectively, show this effect clearly. It can be seen that during more intense geomagnetic activity, the magnitude of the Poynting flux increases in both hemispheres." Can you plot AL on the x-axis and Poynting flux on the y-axis, colour-coded blue/red, to show this more clearly? I agree that the cited figures do show the phenomenon, but I would argue that they are not the best way to make this point.

Lines 118--120: "In such a paradigm, the reduction in nightside Poynting flux observed at Swarm, located below the AAR, is explained as a result of significant energy removed in association with discrete arc auroral electron acceleration above." Are there any satellites above Swarm, or any all-sky imagers below Swarm, that could show this discrete arc?

Lines 150--151: "It can be seen that the sum totals of Poynting fluxes (north hemisphere flux plus south hemisphere flux) remain relatively similar across the seasons on the dayside": Why not plot this sum total on Figure 2 as a purple line, showing this explicitly?

Line 338: Why are the thresholds 25 and 8.75? How were these determined? It would be useful to expand on this, if possible.

Line 356--357: "Average Poynting fluxes for each event are obtained by dividing the integrated Poynting flux values by the event duration." Is this done for each frequency, and is it the average per event per frequency that is used to create the average per frequency seen in Figure 1? i.e., is Figure 1 showing the average Poynting flux of the average Poynting flux of an event, for each frequency? If

so, why not just compute the mean from the Poynting fluxes in each pass at each frequency? If not, then this section needs rewording as I am unsure what the method you're employing is.

Line 358: What are the "three quantities"? The Poynting flux in the three scales, or the magnetic field, electric field, and Poynting flux, or something else? This should be clarified.

Lines 389--390: "Geomagnetic conditions and L1 information can be found at <https://omniweb.gsfc.nasa.gov>." Which data products are used? There are many available at this URL.

Grammar and vocabulary

Line 145: "Significantly" has a specific meaning statistically, I would recommend picking a synonym unless you have a p-value to quote here.

Lines 314--315: Use "Hz" instead of "samples/sec".

Lines 324 and 347: You can either delete "(MLAT)" on both lines or delete "magnetic latitude" on line 347.

“Northern Preference for Terrestrial Electromagnetic Energy Input from Space Weather” by Pakhotin et al. Nature Communications (Manuscript # NCOMMS-20-13230)

Response to Referee 1

We thank the referee for their careful review of our paper, and for the constructive suggestions for how to improve it. We are pleased that the referee finds that in support of our conclusion “that incident Poynting flux at low Earth orbit is on average stronger in the northern hemisphere compared to the southern hemisphere” that we “convincingly argue that this intriguing and surprising finding is geophysical.” The referee did raise some important issues which they wanted to see addressed in order to have confidence that our methods of data analysis supported our conclusions. We very much appreciate these suggestions, and demonstrate below that none of these potential concerns are valid, and our results and conclusions remain sound.

We have now implemented all of the amendments and corrections as suggested. Please note that, as the referee responses have overlapped with the June 2020 release of the latest and most high-quality Thermal Ion Imager dataset to date (Version 0301), we have taken this opportunity to also re-process all of the figures with the new dataset to ensure that our publication uses the best Swarm data which is currently available. This has resulted in some slight changes to the figures; however, the morphology of the results remains unchanged. Most significantly, none of the main conclusions have been affected further demonstrating that our analysis and interpretation remains valid. We hope that with these revisions, the paper will now be considered suitable for publication. Below we address the referee’s comments on a point-by-point basis, with the referee comments in presented in italics and our responses presented in bold.

The paper presents statistical evidence that incident Poynting flux at low Earth orbit is on average stronger in the northern hemisphere compared to the southern hemisphere. The authors convincingly argue that this intriguing and surprising finding is geophysical, and not an effect of measurement bias - although I have one concern in this regard that I would like to see addressed. The authors also give suggestions for what could cause the observed asymmetry. I struggle with the premise of some of these ideas, and I think there are gaps that should be given some attention before the paper is accepted, which I hope it will be. Apart from this, the writing is generally very clear.

Thank you for your interest in our results and for your support of the publication of our manuscript. We have addressed the specific points raised by the referee in detail below, demonstrating that our results are valid. We further also revised the paper to additionally address the gaps raised by the referee.

Detailed review

The potential bias from different orbit orientations relative to the auroral zone is mentioned in the paragraph starting on line 158. The authors seem to be most concerned with the effect this has on the calculation of event averages, which clearly is important. However, I wonder if this impact angle also would affect the calculation of Poynting flux itself, since the ion drift measurements from Swarm are (to my knowledge) less accurate in the along-track direction than cross-track. I would think that, if we performed this analysis with only cross-track data, we would find a northern preference simply because the orbit is more often inclined with respect to the oval in the south, due to the larger pole distance (and since the ion drift is typically zonal). If the observed statistical variation is dominated by the more precise cross-track measurements, this bias could be important. I think you could test it in this way: If you run the analysis mentioned on line 344 (with only cross-track drift/along-track E) with a synthetic constant signal for E, and still get a northern preference, the bias influences the results.

We thank the referee for the suggestion and for bringing this up. Indeed, the relative difference in angles of attack between north and south hemispheres could be an important possible avenue where interhemispheric bias might be unintentionally introduced into the analysis. However, we can demonstrate convincingly that it does not have a significant impact by examining these angles of attack for the whole dataset and for a subset of the auroral zone passes confined to a narrower range of MLT where only +/-1 hour of MLT either side of either noon or midnight was considered. Indeed, when we assessed the angles of attack in both the wider MLT data set, as well as in this narrower and more confined range of MLT, there was no systematic bias in the angles of attack in our data. Significantly, in both the wider and narrower MLT data sets the Northern preference remains. Plots of the angle of attack distributions which demonstrate this are provided below.

In relation to the additional test proposed by the referee, it is not obvious to us how this test of using a constant E-field would reveal a bias in the data set introduced by a difference between the accuracy of the along and cross track electric field data. Indeed, both the magnetic and electric field data can be influenced by the ionospheric conductance so that the effects on both field components need to be assessed in combination. For example, we know from Coxon et al. (2016) that dB variations observed by AMPERE are systematically higher in the northern hemisphere than in the southern hemisphere. Since Poynting flux is calculated from the cross product of residual B and E, and larger dB will likely map to a larger residual B, then this geophysical larger residual B will enter into the Poynting flux calculations.

However, we can consider an alternative test which we hope will satisfy the referee that the northern preference is indeed physical, and does not result from differential bias in the along and cross track electric field components, nor from angle of attack effects. To illustrate this, we can separate the field-aligned Poynting flux into the two components arising from electric fields which are both along track and cross track in the frame of the Swarm orbit. As we show below, the Poynting flux is generated almost exclusively from the more accurate along track electric field (cross track ion drift velocity) and it is this component which when seasonally averaged prefers the North. The less accurate cross track E-field data shows not only a much smaller asymmetry (almost none) but also an extremely small magnitude. This is consistent with the expectation that the majority of the Poynting flux is carried in the quasi-toroidal Alfvén wave polarization with a cross track dB and along track E. Specifically, we reproduced the Poynting flux using only the along-track E-field component ($S = \frac{1}{\mu_0} E_1 \times B_2$), and

also using only the cross-track E-field component ($S = \frac{1}{\mu_0} E_2 \times B_1$). The two scenarios are presented below in Figures 1 and 2, respectively. These features are clearly seen by comparing the two figures (note that the full y scale in Figure 1 is 5 times larger than that in Figure 2).

15 Nov-15 Dec, 2016

1-31 July, 2016

Figure 1: Poynting flux statistics recalculated using the along-track E-field only (same format as Figure 1 in the main manuscript)

Figure 2: Poynting flux statistics recalculated using the cross-track E -field only (same format as Figure 1 in the main manuscript)

It can be seen that, in agreement with the expectations you have raised in the referees comments, most of the energy appears to indeed be in the E_1 and B_2 component, i.e. the cross-track ion drift component.

To calculate the angle of attack distributions, the impact angle was calculated as the variation from beginning to end of each event of x and y (in Cartesian coordinates) transformed from a magnetic latitude (MLat) by magnetic local time (MLT) grid (the same map as was displayed in Supplementary Material Figure 6 in the manuscript. Here the MLT values are multiplied by 15 to map them onto the 0-360 degree circle range). This was done for each of the four different seasonal time periods, and the

results are displayed in Figures 3 through 6 below. In each figure, panels (a) and (b) denote the north and south hemisphere distributions, respectively, for events limited to ± 1 hour MLT on either side of local noon or midnight. Meanwhile panels (c) and (d) display the distributions for all MLT ranges. Overall, there is no obvious systematic bias in the angles of attack in either the narrow or wider MLT bins.

Figure 3: impact angle histogram distributions for the north hemisphere (a, c) and south hemisphere (b, d) for MLTs restricted to ± 1 hour within either noon or midnight (a, b), as well as for all MLTs (c, d), for the time period 15 Nov-15 Dec, 2016.

20 Feb-25 Mar, 2016

+1 hour MLT around noon or midnight

Figure 4: impact angle histogram distributions for the north hemisphere (a, c) and south hemisphere (b, d) for MLTs restricted to +/-1 hour within either noon or midnight (a, b), as well as for all MLTs (c, d), for the time period 20 Feb-25 Mar, 2016.

20 March-20 April, 2017

+1 hour MLT around noon or midnight

Figure 5: impact angle histogram distributions for the north hemisphere (a, c) and south hemisphere (b, d) for MLTs restricted to +/-1 hour within either noon or midnight (a, b), as well as for all MLTs (c, d), for the time period 20 Mar-20 Apr, 2017.

1-31 July, 2016

+1 hour MLT around noon or midnight

Figure 6: impact angle histogram distributions for the north hemisphere (a, c) and south hemisphere (b, d) for MLTs restricted to ± 1 hour within either noon or midnight (a, b), as well as for all MLTs (c, d), for the time period 1-31 July, 2016.

It can be seen that in each case the distributions for the north and south hemisphere are very similar, and mostly limited to 0-30 degrees. As such, any theoretical bias due to angle of attack is expected to affect the results in both hemispheres equally, since the distribution of angles of attack are very similar in both hemispheres.

With this we believe we may reasonably conclude that any angle of attack effects, if they occur, seem to affect both hemispheres approximately equally, so we can conclude that no significant systematic bias is caused by angle of attack considerations.

In the discussion (paragraphs starting on line 216), I get the impression that the magnetosphere radiates equal amounts of energy to the two hemispheres (e.g., the use of "fixed equatorial energy source"), and that it gets partitioned and reflected in different ways near the ionosphere. For example, it is claimed that since the Southern hemisphere can be more dark, more energy is being diverted into accelerating electrons at the expense of Alfvén wave power, and therefore wave power is stronger in the north. If this is intended, I think the premise of symmetric incident energy should be justified. I see no reason why it should be symmetric to begin with, since different conductivities imply that different amounts of energy are required to bring the ionosphere and magnetosphere in balance. To first order I would expect more Alfvén waves going back and forth between magnetosphere and ionosphere in the most sunlit hemisphere, simply because that hemisphere is harder to move due to coupling with neutrals. "First order" because particle precipitation and feedback instabilities complicate the picture.

We thank the referee for this comment. We certainly agree that there could be asymmetries present in the directionality of the energy flow from the magnetosphere to the ionosphere. Indeed, it was not our intention to imply that our results require an equipartition of energy into both hemispheres. In fact, our main point is that back-coupling from the ionosphere to the magnetosphere seems to be of critical importance. As such, in our view, any studies of how the magnetosphere drives the ionosphere needs to include as an essential element the impacts arising from the back reaction of the ionosphere on the magnetosphere. Our results suggest that this strong M-I coupling plays a key role in particular in creating the interhemispheric asymmetry in energy flow we report here and that does not seasonally average to zero.

If the ionosphere were a passive rather than an active component of magnetosphere-ionosphere coupling, we would expect any magnetospheric asymmetries to statistically reverse 6 months later such that the seasonally averaged energy input would be expected to be symmetric into both hemispheres. The fact that this does not happen, and that even when seasonally averaged, northern energetics are higher, suggests an active role for the ionosphere in redistributing energy. On the dayside, we postulate that this could be due to the fact that, although the amount of sunlight falling on each hemisphere is symmetric when seasonally averaged, the amount of sunlit auroral oval is not, due to the relatively greater relative offset of the southern oval from the rotation axis. Similar asymmetries in the magnetic fields in both hemispheres may be important on the nightside as well, for reasons discussed in the manuscript.

The reference in the manuscript to a 'fixed equatorial energy source' was related to the fact that on the dayside, it appears that the sum total of energies in both hemispheres (blue + red plots in Figure 2) appear to add together to give a constant sum across different seasons (a new purple line in Figure 2 has been added to show this summation).

The argument that Alfvén wave power is lost to particles seems to imply a prediction: If you measure Alfvén wave power in or above the AAR, you should get a different result in terms of north-south asymmetries, since the partition has not yet happened. There is at least one paper that has used FAST to statistically analyze Alfvén wave power, separately in the two hemispheres (Hatch et al., 2018,

<https://doi.org/10.1016/j.jastp.2017.08.009>). It does not seem to be directly comparable, but it may be worth mentioning.

We completely agree and hope our study will be used in conjunction with studies above the AAR such as that mentioned by the referee to further divulge what happens within the AAR and how this behavior plays out across different hemispheres. We believe that Swarm data in the context of our study represents residual energy downstream of AAR processes and so this study must be used in conjunction with studies using datasets such as e.g. Polar, Cluster etc. to place the findings in the proper wider context.

As far as the Hatch et al. (2018) study, we completely agree that it is of relevance and thank the referee for highlighting it. It has been added into the reference list and a new sentence added to the paper at line 279 “Interestingly, the results of [27], using statistics obtained from the FAST satellite, which traverses a range of heights above Swarm, also lend some support to the above hypothesis.”. That study was conducted primarily to study energy fractions as a function of storm phase, using data from FAST whose orbit is elliptical with the apogee passing close to the auroral acceleration region and the perigee below it. As such it is unclear from their results which of the observations were made at or near AAR and which were made below it in regions closer to Swarm altitudes (results at heights below 750 km are also discounted in the Hatch et al. study). In addition, although their study provides Poynting flux numbers for both hemispheres it does not explicitly separate between day and night, nor height. The paper also declares explicitly that southern hemisphere numbers are subject to greater uncertainty.

On line 290, it is claimed that this result can have implications for understanding long-term atmospheric climate response. This seems very speculative, and I recommend removing it unless further justification and citations can be given.

Thank you for the suggestion, we have amended this sentence and removed references to atmospheric climate response as per your recommendation.

Minor issues:

MIC: I think it would be better to spell out magnetosphere-ionosphere coupling.

Thank you, all instances of MIC have been spelled out as “magnetosphere-ionosphere coupling” as per your recommendation

line 89: "energy transfer remains slightly northward" - Do you really mean to say that the median direction is northward?

Thank you for flagging up this ambiguity, this has now been corrected to: “but remarkably the northern preference in Poynting flux appears to persist even near the northern winter solstice on the nightside.”

line 91: Missing "to" before "be"

Thank you, this has been amended.

figure 2: I strongly suggest using the same color scheme in the two columns

Thank you for your suggestion, this has been modified.

line 222: remove "of" before "shown"

Thank you, this has been amended.

line 258: The only way I can make sense of this sentence is if the second "both" is removed.

Thank you, this has been amended.

line 358: "these three quantities" which?

Thank you for flagging this up, indeed this is unclear language. It has been amended from "these three quantities" to "Poynting flux values".

Response to Referee 2

We thank the referee for their careful review of our paper, and for the constructive suggestions for how to improve it. We are pleased that the referee "*enjoyed this manuscript and feel[s] it would be suitable for publication in Nature*". We have taken their valuable comments and suggestions into consideration as we revised our manuscript, and believe that we have addressed all of their concerns. We hope that with these changes you can now recommend our manuscript for publication in Nature Communications.

Please note also that, as the revision overlapped with a new dataset release for the Thermal Ion Imager (version 0301 was released in June 2020), we have taken this opportunity to also re-process all of our figures incorporating this latest and most accurate dataset. The use of the newest data ensures that our results are supported by the best available data. This resulted in some slight changes to the figures, but none of the core conclusions of the manuscript were impacted.

We further address the referee comments point-by-point below. The referee comments are presented in italics, while our responses are presented in bold.

In general I enjoyed this manuscript and feel it would be suitable for publication in Nature with some minor changes, which I enumerate below. My most serious criticism is of the method, particularly lines 347–359; I did not feel this went into enough detail to aid reproducibility.

Thank you. We address the concerns with the methodology in detail below. This additional analysis further validates that our conclusions are appropriate and valid.

Science

Line 81–84: This is very similar to the result of [12], which may be worth noting here.

Thank you for your comment. For greater clarity and in consideration of your suggestion, we have amended the text first citing [12] so it now reads: "Interestingly, recent work found that FACs in the auroral zone tend to be stronger in the north when averaged over a year [12]." (The underlined section shows the new addition). The difference of course is that FACs alone do not directly represent energy flow while Poynting flux does. As such we believe our results effectively complement the findings of [12] to demonstrate that the observations made by the authors of that publication are also representative of the result we present here that there is a significant asymmetry in magnetosphere-ionosphere coupling energetics.

Figure 1: Is there a reason not to show similar pair(s) of panels for the spring and/or autumn equinox?

Thank you for this comment. The original reason of the omission was simply for the purposes of brevity, since the near-equinox asymmetries are already demonstrated for small, meso- and large scales in Figure 2. However, as the referee has indicated that readers may be interested in these additional figures, we have now added them to Figure 1.

Lines 96--97: "Supplementary Material Figures 1 and 4, for the northern near-winter and near-summer solstice periods, respectively, show this effect clearly. It can be seen that during more intense geomagnetic activity, the magnitude of the Poynting flux increases in both hemispheres." Can you plot AL on the x-axis and Poynting flux on the y-axis, colour-coded blue/red, to show this more clearly? I agree that the cited figures do show the phenomenon, but I would argue that they are not the best way to make this point.

Thank you for the suggestion. We have now added the AL Index vs Poynting flux scatter plots, as well as lines of best fit, to the Supplementary Material Figures 1-4 as per your suggestion.

Lines 118--120: "In such a paradigm, the reduction in nightside Poynting flux observed at Swarm, located below the AAR, is explained as a result of significant energy removed in association with discrete arc auroral electron acceleration above." Are there any satellites above Swarm, or any all-sky imagers below Swarm, that could show this discrete arc?

We of course agree this is a very interesting point. We hope and anticipate that the publication of our paper will lead to follow-on work to test the hypothesis that more discrete arcs would be expected in the southern nightside hemisphere. For example the e-POP satellite with its Fast Auroral Imager (FAI) experiment (Cogger et al., 2014) is able to resolve discrete arcs, as reported in e.g. Miles et al. (2018). A number of other international mission proposals have also been developed which focus on the interhemispheric nature of discrete auroral arcs. We believe that this would be a very valuable follow-on study and intend to look into carrying it out as part of future work in this direction. The THEMIS All-Sky Imager (ASI) network may also be used for discrete arc detection, however they are located only in the northern hemisphere and so direct conjugate comparisons of the southern hemisphere may prove difficult. Nonetheless, we believe that an examination of the discrete aurora would be a natural extension of the work we present here.

Lines 150--151: "It can be seen that the sum totals of Poynting fluxes (north hemisphere flux plus south hemisphere flux) remain relatively similar across the seasons on the dayside": Why not plot this sum total on Figure 2 as a purple line, showing this explicitly?

Thank you for your suggestion, this has now been added to Figure 2.

Line 338: Why are the thresholds 25 and 8.75? How were these determined? It would be useful to expand on this, if possible.

Thank you for your question. The two thresholds were determined empirically on a trial-and-error basis. If the threshold is too high, the filter rejects many events, meaning statistics could become less reliable with smaller sample sizes. On the other hand, if thresholds are set too low, the filter may introduce events which are mostly characterized by noise rather than a meaningful signal. The thresholds are thus selected to strike a balance between capturing meaningful signal and obtaining reasonable case counts so that the sample size forms a well-behaved distribution that is amenable to reliable statistical analysis.

Line 356--357: "Average Poynting fluxes for each event are obtained by dividing the integrated Poynting flux values by the event duration." Is this done for each frequency, and is it the average per event per frequency that is used to create the average per frequency seen in Figure 1? i.e., is Figure 1 showing the average Poynting flux of the average Poynting flux of an event, for each frequency? If so, why not just compute the mean from the Poynting fluxes in each pass at each frequency? If not, then this section needs rewording as I am unsure what the method you're employing is.

Thank you for your question. For each event, and for each of the low-pass filtered time series, the time-integrated Poynting flux is divided by event duration to get the average Poynting flux along the Swarm track during the event. Such a measure allows the intercomparison of events with different overall temporal durations, and demonstrates the time averaged amount of Poynting flux remaining after the application of the time domain filter. Note that since the filters are applied in the time domain, there is explicit calculation of a per frequency Poynting Flux. We choose this time domain approach in an attempt to reflect the characteristics of the wavepackets observed. Hence the only "averaging" which is applied is through a division of the time-intergrated Poynting flux along the spacecraft track by the duration of the event.

The median (not the mean!) of these average Poynting fluxes is then used to form the plots in Figures 1 and 2, with the error bars representing the quartiles of the distribution. We use the median as opposed to the mean since we do not expect the Poynting flux statistics to follow normal distributions, and the median is expected to more representative of the behavior of such distributions. We hope this explanation further clarifies our methodology.

Line 358: What are the "three quantities"? The Poynting flux in the three scales, or the magnetic field, electric field, and Poynting flux, or something else? This should be clarified.

Thank you for this question. Indeed this is unclear language. As mentioned in the reply to Referee 1, this has now been re-worded from "these three quantities" to "Poynting flux values".

Lines 389--390: "Geomagnetic conditions and L1 information can be found at <https://omniweb.gsfc.nasa.gov>." Which data products are used? There are many available at this URL.

Thank you for bringing this up. For greater clarification, the sentence has been amended to: "Geomagnetic conditions and L1 information (IMF Bx, IMF By, IMF Bz, AE index, AL index) can be found at <https://omniweb.gsfc.nasa.gov>."

Grammar and vocabulary

Line 145: "Significantly" has a specific meaning statistically, I would recommend picking a synonym unless you have a p-value to quote here.

Agreed and amended, thank you.

Lines 314--315: Use "Hz" instead of "samples/sec".

Thank you for bringing this up. We use samples/sec here to avoid confusion as it can be taken by some readers to mean either 'sampled x times per second' or 'the sampling frequency is x Hz' when in reality the Nyquist frequency would be x/2 Hz. The 'samples/sec' notation is a direct quote from the original instrument paper by Knudsen et al. (2017) where we believe the authors deliberately used this form to avoid any misunderstandings.

Lines 324 and 347: You can either delete "(MLAT)" on both lines or delete "magnetic latitude" on line 347.

Agreed and amended, thank you.

REVIEWERS' COMMENTS

Reviewer #1 (Remarks to the Author):

I have read the updated manuscript and the response letter to the reviewers, and find that the authors have thoroughly addressed the concerns that were raised. I recommend that the paper is published as is.

Reviewer #2 (Remarks to the Author):

In general I am satisfied with the authors' responses and would recommend this manuscript for publication in Nature Communications.

I have a minor comment, which is that I try to avoid using the word "average" in technical manuscripts if at all possible. During the proofing stage, I would recommend swapping this for "median" (or "mean") throughout, as the authors' response to my comments made it clear that I had misinterpreted their intent at least once during my read of the paper, and I am concerned that this misinterpretation may also occur in the manuscript's readers.

All in all, however, I am happy for this to be published without seeing it for review a third time.

We thank both referees for their review of the resubmission and as Referee 1 did not have any further comments, we address the minor comment raised by Referee 2 below. Our response is in bold.

Response to Referee 2

“In general I am satisfied with the authors' responses and would recommend this manuscript for publication in Nature Communications.

I have a minor comment, which is that I try to avoid using the word "average" in technical manuscripts if at all possible. During the proofing stage, I would recommend swapping this for "median" (or "mean") throughout, as the authors' response to my comments made it clear that I had misinterpreted their intent at least once during my read of the paper, and I am concerned that this misinterpretation may also occur in the manuscript's readers.

All in all, however, I am happy for this to be published without seeing it for review a third time.”

We thank Referee 2 for raising this issue and completely agree. In particular we recognize that in the Methods section since we both use mean and median, it is important to clarify which method is utilized where. As such we have replaced ‘average’ with ‘mean’ or ‘median’ as appropriate. The only other cases where the word ‘average’ is used is in the sense of ‘averaged over seasons’, ‘averaged over a year’ etc. In this case the term is used to mean that seasonal variations have been factored out. We believe that in this sense the word ‘average’ is the best way to convey the meaning in the English language and do not wish to add more uncertainty by re-writing it in potentially more ambiguous terms. As such we have decided to retain ‘average’ only where it refers to averaging over seasonal variations, as in, to imply factoring out seasonal variations.